# A Hemodynamic Pulse Wave Simulator Designed for Calibration of Local Pulse Wave Velocities Measurement for Cuffless Techniques

**DOI:** 10.3390/mi14061218

**Published:** 2023-06-09

**Authors:** Cheng-Yan Guo, Jau-Woei Perng, Li-Ching Chen, Tung-Li Hsieh

**Affiliations:** 1Accurate Meditech Inc., New Taipei City 241406, Taiwan; gcy626@gmail.com; 2Department of Mechanical and Electromechanical Engineering, National Sun Yat-sen University, 70 Lienhai Road, Kaohsiung 80424, Taiwan; jwperng@faculty.nsysu.edu.tw; 3LAICA International Corp, New Taipei City 231, Taiwan; p10e43027@ntu.edu.tw; 4Department of Electronics Engineering, National Kaohsiung University of Science and Technology, Kaohsiung 80778, Taiwan

**Keywords:** cuffless blood pressure, blood pressure monitoring, pulse wave velocity, multiple linear regression, mean absolute error

## Abstract

Objective: Devices for cuffless blood pressure (BP) measurement have become increasingly widespread in recent years. Non-invasive continuous BP monitor (BPM) devices can diagnose potential hypertensive patients at an early stage; however, these cuffless BPMs require more reliable pulse wave simulation equipment and verification methods. Therefore, we propose a device to simulate human pulse wave signals that can test the accuracy of cuffless BPM devices using pulse wave velocity (PWV). Methods: We design and develop a simulator capable of simulating human pulse waves comprising an electromechanical system to simulate the circulatory system and an arm model-embedded arterial phantom. These parts form a pulse wave simulator with hemodynamic characteristics. We use a cuffless device for measuring local PWV as the device under test to measure the PWV of the pulse wave simulator. We then use a hemodynamic model to fit the cuffless BPM and pulse wave simulator results; this model can rapidly calibrate the cuffless BPM’s hemodynamic measurement performance. Results: We first used multiple linear regression (MLR) to generate a cuffless BPM calibration model and then investigated differences between the measured PWV with and without MLR model calibration. The mean absolute error of the studied cuffless BPM without the MLR model is 0.77 m/s, which improves to 0.06 m/s when using the model for calibration. The measurement error of the cuffless BPM at BPs of 100–180 mmHg is 1.7–5.99 mmHg before calibration, which decreases to 0.14–0.48 mmHg after calibration. Conclusion: This study proposes a design of a pulse wave simulator based on hemodynamic characteristics and provides a standard performance verification method for cuffless BPMs that requires only MLR modeling on the cuffless BPM and pulse wave simulator. The pulse wave simulator proposed in this study can be used to quantitively assess the performance of cuffless BPMs. The proposed pulse wave simulator is suitable for mass production for the verification of cuffless BPMs. As cuffless BPMs become increasingly widespread, this study can provide performance testing standards for cuffless devices.

## 1. Introduction

Hypertension is a chronic disease that does not pose any immediate danger to the patient; however, the presence of hypertension can lead to the development of many diseases. Hypertension must be diagnosed based on a patient’s 24 h blood pressure (BP) change [1], so a non-invasive electronic BP monitor (BPM) must be used that can continuously monitor BP. This type of BPM is called an ambulatory BPM (ABPM). The ABPM measures BP by intermittently inflating a cuff and pressurizing the patient’s arm during diagnosis, often every 30 min or hourly. However, this diagnostic method can affect the daily life of the patient and may cause uncertainty in the measurement accuracy [2]. To improve the time resolution and accuracy of ABPM measurement, cuffless BP monitoring technologies have been proposed. Any BPM without an occluding cuff can be described as cuffless technology [3]. Cuffless BPMs based on pulse arrival time (PAT) measure the propagation time of the PAT through electrocardiography and photoplethysmography sensors [4]; regression is then performed on the PAT and BP values measured using a reference sphygmomanometer to obtain a correction coefficient [5]. However, some studies have observed that the PAT will be affected by the pre-ejection period [6,7], making PAT an inaccurate estimate of BP. Therefore, in recent years, studies of cuffless technology have begun to use other technical solutions. One is to measure the time difference generated by the pulse waves at two different positions on the local artery. This time difference is called the local pulse transit time (PTT). The pulse wave is caused by the heart’s pressure, which causes the arterial wall to expand acutely. Therefore, measurements of an arterial segment’s PTT can be used to estimate the artery’s BP [8]. Current sensors used to measure PTT include piezoelectric material-based pulse wave sensors [9]. In this approach, high-sensitivity sensors based on elastomer-based strain gauges are placed on the subjects’ skin [10]. The arterial pulsometer is made using micro-electromechanical systems technology and pressure pulse wave sensors made from piezoresistive elements [11]. These compact sensors can be integrated into wearable devices to locally measure the subject’s arteries, such as the carotid artery or the wrist’s radial artery, from which PTT can be easily measured. The BP estimation principle using PTT measurement technology involves converting the PTT into pulse wave velocity (PWV) based on the distance between two sensors at different positions and then using hemodynamics to correlate the PWV with BP.

However, current PTT-based cuffless BPMs require effective and reliable verification methods for industrial production that can ensure the measurement consistency of each cuffless BPM [12]. Traditional BPMs use an oscillometric method based on the cuff’s air pressure change. Factory calibration of this type of equipment can be performed using an air pressure simulator to simulate a range of BP and pulse rate values to ensure consistency throughout the mass production process. This approach is also the standard safety testing and performance verification method adopted by regulatory agencies worldwide [13]. Compared with air pressure simulators used for traditional BPMs, simulators for cuffless BPMs are relatively challenging to design for two reasons. First, there is currently no general measurement technology and method for cuffless BPMs, leading to differences in validation methods for cuffless BPMs due to measurement technology differences. Second, simulators based on the hemodynamic BP estimation method must consider more parameters than traditional BPM air pressure simulators, including the tube diameter, tube wall, fluid density, and pressure generated by the water pump [14,15]. Therefore, in this study, we propose a design for a cuffless BPM simulator for local PTT and demonstrate a method to calibrate the deviation of the BP curve measured by the cuffless BPM. This method can be used to quantify the deviation of the same type of cuffless BPM and rapidly calibrate this equipment.

### Our Contribution

This study suggests a method to develop a pulse wave simulator capable of simulating the human cardiovascular system in vitro including complex hemodynamic phenomena. In addition, we design a cuffless BPM for local PTT measurement, which is used to measure the pulse wave simulator’s PWV. Based on the principle of hemodynamics, we discuss the simulated pulse waves generated in this research; this information is then used in the calibration method for the cuffless BPM. This paper’s primary contribution is to show how a pulse wave simulator can be used to efficiently correct the bias of BP measurements from a cuffless BPM.

## 2. Materials and Methods

In this study, a pulse wave simulator is designed that can be used to verify cuffless BPMs using local PWV. This pulse wave simulator must be able to simulate the characteristics of hemodynamics, including simulating the ejection of blood from the heart and the circulation of blood through the arteries. In addition, the arteries’ mechanical properties must be able to transmit the pulse wave’s properties [16]. Therefore, a range of technical details should be considered in the design of the pulse wave simulator, including the electromechanical system used to simulate the regular heart rhythm. In the proposed design, the arm’s skin tissue phantom uses silicone rubber. Silicone rubber tubing has similar properties to the radial arteries and is thus used as an arterial phantom for fluid circulation in the pulse wave simulator. An expanded polyethylene foam (EPE) adult arm-size model is constructed, and the arterial phantom is embedded into the model, with a skin phantom then used to cover the resulting arm model.

### 2.1. Design of Hemodynamic Simulator

The pulse wave simulator designed in this study simulates cardiovascular circulation in vitro. The design uses elastic tubes embedded in a pulse simulator model to simulate the physiological phenomenon of pulse wave transit during human circulation [17]. The physiological principle of diastolic blood pressure (DBP) is determined by peripheral vascular resistance; in contrast, systolic blood pressure (SBP) is directly determined by left ventricular ejection, arterial stiffness, and pulse wave reflection.

### 2.2. System Architecture of Hemodynamic Simulator

Figure 1 shows the system diagram for the pulse wave simulator proposed in this study. The water reservoir stores the water in a simulated circulatory system. An inflatable bulb pump and aneroid sphygmomanometer (Yuyue Medical, Jiangsu, China) are connected to the water reservoir. The inflatable bulb pump is used to adjust the base pressure (P_0_) of the pulse wave simulation system. The aneroid gauge can then be used to confirm whether the basic pressure of the simulator has reached the target value. An electric pump (JT-750, Jovtop Electronics Co., Quanzhou, China) is used as the heart in the simulator, pumping the water through the elastic tube that forms the simulated circulatory system. The pressure of the water increases after passing through the electric pump. The electric pump makes a pressure change, referred to as the pulse pressure (PP), where the peak pressure is the SBP. If the frequency of the electric pump is set to 1 Hz, a heart rhythm of 60 beats per minute can be simulated. A pressure sensor (MPS-3117, Metrodyne Microsystem Crop., Hsinchu, Taiwan) is used to monitor the simulator’s pressure change signal. This signal is similar to the invasive arterial BP waveform. Continuous monitoring of this signal can confirm the simulator’s SBP, DBP, and mean arterial pressure (MAP). Before using the simulator, regression calibration must be performed on the pressure sensor and aneroid gauge.

Figure 2 shows our designed pulse wave simulator. Figure 2a shows an adult-size arm model (NM2255, Kangway Medical Co., Shanghai, China) that can wear the cuffless BPM for testing. In this design, the elastic tube is embedded in the arm model at a depth of 3 mm. The elastic tube is made of silicone rubber (Silicone Rubber #3428, Emperor Chemical Co., Taipei, Taiwan), and the Shore hardness of the silicon rubber is 28 A, with an inner diameter of 4 mm and a wall thickness of 1 mm. The tube’s properties will affect the backward wave’s reflection delay, such as length, diameter, and wall thickness [18,19]. Moreover, since arterial bifurcation, plaques, and terminal arterioles will affect the reflected wave, the reaction force generates a wave [20]. This wave will move backward (or centrifugally) with the same velocity, and the transverse displacement of the centripetal wave is reversed or 180°. Then, the position of the cuffless BPM set in this simulator also affects the delay of the measured reflection wave. In this study, we placed the cuffless BPM at 2 cm before the tube turn, similar to placing the device at 2 cm above the styloid process when testing the wrist of a real person [21]. Figure 2b shows the system’s aneroid gauge. When using the inflatable bulb pump shown in Figure 2d to set the base pressure, the pressure value displayed on the aneroid gauge can be used to confirm the current pressure of the simulator. Figure 2c shows the printed circuit board (PCB) of the simulator’s control system, the electric pump, and the water reservoir. The elastic tube embedded in the arm model must be connected to the electric pump and water reservoir to form a simulated circulatory system; this elastic tube must also be connected to the system’s pressure sensor. The connection between the pressure sensor and the elastic tube must form a barrier to water and air; this is achieved using fluoroplastic porous film (ST2026AN06A1, Nitto Denko Co., Osaka, Japan). The level of all the components that comprise the pulse wave simulator must be within 10 cm of the elevation of the water surface in the water reservoir to avoid hydrostatic effects that would cause the pressure in the elastic tube to differ from that of the pressure sensor [22,23].

Figure 3 shows the control system combining the simulator’s electric pump and water reservoir. The electric pump in Figure 3a is used to pump water into the simulated circulatory system to simulate the action of the heart. Note that the inlet is configured to suck water from the reservoir into the pump, while the outlet discharges water to the elastic tube embedded in the arm model, which then returns to the reservoir after circulation. Figure 3b shows the control system’s PCB. The main components of the PCB include a pressure sensor and motor driver (RZ7899, Smart Microelectronics Co., Wuxi, China) to control the electric pump and microcontroller (STM32F030F4P6, STMicroelectronics Inc., Geneva, Switzerland). This PCB can adjust the pulse wave signal period according to a specified heart rate and pulse wave magnitude and correspondingly change the pulse-width modulation (PWM) signal output to the motor driver. Figure 3c shows the water reservoir used to store water in the simulated circulation system. It should be noted that the water reservoir requires O-ring seals to maintain the base pressure of the pulse wave simulation system.

### 2.3. Characteristic of Hemodynamic Simulator

Based on the system design of our simulator, the corresponding pressure signal waveform of the pulse wave simulator is shown in Figure 4. When the electric pump pumps water into the elastic tube, the pressure in the tube will increase, and the pressure waveform will generate a peak. This peak value represents the SBP, while the trough when the pressure decreases to its minimum represents the DBP. The difference between the SBP and DBP is the PP, as shown in Equation (1). The PP is caused by the difference in the pulse magnitude of the electric pump pumping water into the elastic tube caused by the different PWM duty cycles. The MAP is approximately equal to the sum of 1/3 of the SBP and 2/3 of the DBP, as shown in Equation (2) [24]. However, for ease of calculation, we use the pressure sensor’s root-mean-square (RMS) value per second as the reference MAP value during the experiment.
(1)PP=SBP−DBP
(2)$$\mathrm{MAP}\approx \frac{\mathrm{SBP}}{3}+\frac{2\cdot \mathrm{DBP}}{3}$$

When the electric pump of the pulse wave simulator pumps water into the elastic tube, different PP and SBP values can be achieved based on the different PWM duty cycles, as shown in Figure 5. When the power of the electric pump increases, the PP and SBP also increase, while the DBP remains close to the base pressure value set point. Since the pressure change of the pulse wave simulator is based on the PP generated by the electric pump, the MAP values are also affected; thus, the duty cycle of the PWM is positively correlated with PP, SBP, and MAP. In addition, if used to verify the BP validation ranges required by ANSI/AAMI/ISO 81060-2 standards, the range of SBP is 100 mmHg to 160 mmHg, and the range of DBP is 60 mmHg to 100 mmHg [25,26]. The pulse wave simulator proposed in this study can control the simulated pressure range by adjusting the base pressure and the electric pump’s power.

Based on the hemodynamic principle [27], in cuffless BPM research, equations can be used such as the Bramwell–Hill (BH) equation in Equation (3) and the Moens–Korteweg (MK) equation in Equations (4) and (5) [28]. The modified BH model can be expressed as a function of ΔP relative to the change in arterial cross-sectional area, where ΔP is the PP, V is the initial volume, ΔV is the volume change, and ρ is the blood density, where the whole blood density of 1.06 g/mL is typically used [29]. Given the solution used in this study, the density of water was instead used, and the BH model is derived from the MK equation. In the MK equation, D is the diameter of the arterial phantom, h is the wall thickness of the arterial phantom, and ρ is the density of the water used in the simulator. E is the Young’s modulus value of the arterial phantom [30], E0 is the elastic modulus at zero pressure, and γ is the coefficient depending on the particular vessel of the Young’s modulus parameter. The average values obtained in previous studies’ trials performed on the brachial artery were 1428.7 and 0.031 for these two parameters, respectively [31,32]. P is the pressure inside the arterial phantom, which is close to the MAP [33]. Therefore, based on the MK equation, in theory, the proposed simulator can simulate the PWV range shown in Figure 6 given that the electric pump’s different PWM duty cycles can control the PWV levels, and the PWV changes are a non-linear function based on the hemodynamic principle.
(3)$$\mathrm{PWV}=\sqrt{\frac{V\cdot \mathrm{dP}}{\rho \cdot \mathrm{dV}}}$$
(4)$$\mathrm{PWV}=\sqrt{\frac{E\cdot h}{D\cdot \rho }}$$
(5)$$E=E_{0}\cdot \exp^{\gamma \cdot P}$$

### 2.4. Experiment Setup of Validation for Pulse Wave Velocity

In this study, we use a cuffless BPM (Accurate 24 BPM, Accurate Meditech Inc., New Taipei, Taiwan) based on local PWV measurement as the device under test (DUT) [34] with our proposed pulse wave simulator. As shown in Figure 7, the pulse wave simulator must be started first, and the position of the elastic tube embedded in the arm phantom must be confirmed through palpation. After the position of the arterial phantom is confirmed, the cuffless BPM is worn on the arm model. The cuffless BPM, as the DUT, has two piezoelectric ceramic pulse wave sensors, as shown in Figure 8. The microprocessor samples the pulse wave signal of the DUT’s sensor through the analog front-end. The cut-off frequencies of the high-pass and low-pass filters of the analog front-end of the DUT are 0.6 Hz to 10.6 Hz, respectively. After sampling the sensor’s analog signal into a digital signal, the finite impulse response filter is used for the pulse wave’s digital signal filtering, and the cut-off frequency of the digital filter is 0.7 Hz to 9.5 Hz, respectively [35]. After signal processing, the dynamic feature detection algorithm in our previous research can effectively detect the first peak of the pulse wave [36]. The gradient of the second peak caused by the reflected wave is smaller than that of the first, so our algorithm will not detect it. The PTT can be calculated by the time difference between the peaks detected by the two sensors. At a sampling rate of 5 kHz, the PTT measurement resolution can reach ±0.2 ms. The PTT can be converted to PWV using Equation (6), where the sensor distance (L) of the DUT is 4 cm. Based on the theory of the MK equation, the range of PWV and PTT values that the pulse wave simulator can produce are shown in Figure 9, and the theoretical range that can be measured by the DUT lies within the red dashed rectangle. It should be noted that the PWV and PTT are inversely related. In the experimental results, this study uses the PWV as a parameter to measure the performance of the pulse wave simulator by the DUT.
(6)$$\mathrm{PWV}=\frac{L}{\mathrm{PTT}}$$

## 3. Results and Discussion

### 3.1. Experiment Analysis

Before the cuffless BPM is investigated as a DUT, we must first model the PWV of the pulse wave simulator. Based on the MK equation, the relationship between PWV and BP is a nonlinear function. However, in practice, the hemodynamic characteristic curve of the pulse wave simulator does not conform to theoretical predictions, hence, the reference equipment must be used for calibration. The reference equipment for the calibration model can be the same type of cuffless BPM or a similar sensor. This study uses a piezoelectric sensor (HK1205, Huake Electronic Co., Hefei, China) with the same sensing principles as the cuffless BPM (i.e., the DUT) for local PWV measurement, and the measured PWV is used as a pre-calibration reference value. As shown in Figure 10, there is a bias between the reference PWV values and the theoretical values estimated by the MK equation. Traditionally, a multi-parameter calibration function can be established through an MLR model [37], and multiple variables can establish a mapping function that maps the PWV measured by cuffless BPM to the MK equation [38,39,40]. The MLR model for calibration is shown in Equation (7), where PWV_ref_ is the reference value of PWV, P_ref_ is the pressure value measured by the pressure sensor of the pulse wave simulator, β_0_ is the intercept term, and β_1_ and β_2_ are coefficient terms. After calibration using the MLR, the PWV reference values can be effectively mapped to BP using the MK equation. We hence use the MLR model to calibrate the cuffless BPM DUT values.
(7)$$c=\beta_{0}+\left(\beta_{1}\cdot P_{\mathrm{ref}}+\beta_{2}\cdot {\mathrm{PWV}}_{\mathrm{ref}}\right)$$

We then used the cuffless BPM as the DUT to measure the pulse wave simulator, with the simulator set to 75 beats per minute, corresponding to a simulated pulse wave frequency of 1.25 Hz. We placed the DUT in the pulse wave simulator and adjusted the duty cycles to simulate BP changes. Nine different simulated BP set points were investigated, with a measurement time of 60 s for each set point; thus, the sampled data for each set point contain 75 PWV values. In the experiment, we divided the DUT measurements into PWV values obtained by direct measurement and calibrated PWV values after applying the MLR model calibration. As shown in Figure 11, the PWV data sampled at each set point are presented in the form of a box plot. In the data without MLR model calibration, the mean absolute error (MAE) of the PWV for the theoretical values derived using the MK model is 0.77 m/s; in contrast, using the MLR model to calibrate the DUT measurements significantly improved the MAE to 0.06 m/s. After calculating the median PWV sampled at each set point in the two measurement experiments, we used Bland–Altman plots to compare the model results. The standard deviation values of the experiments without MLR model calibration (Figure 12) and those calibrated with the MLR model (Figure 13) were ±0.11 mmHg and ±0.06 mmHg, respectively. Thus, using the MLR model to calibrate the data obtained from the cuffless BPM in this study does not adversely affect the PWV sampling [41]. A significant difference is recorded in terms of the mean PWV measurement accuracy, with mean differences of 0.78 mmHg and −0.05 mmHg, respectively, recorded for the measurements without and with MLR model correction. In the BP value range of 100–180 mmHg, the observed PWV measurement deviations will cause measurement errors of 1.7–5.99 mmHg before calibration and 0.14–0.48 mmHg after calibration using the MLR model. The measurement errors using the MK model will have a greater impact on the BP accuracy when the PWV is slower.

Based on our proposed calibration method and experimental results, the MLR model can effectively be mapped to the theoretical curve derived from the MK model of the pulse wave simulator. This calibration can then be used to validate mass-produced cuffless BPM devices to quantitative standards. Since the MLR model will map the measured values of the cuffless BPM to the theoretical values of the MK model, if the biases of the values measured by the cuffless BPM are significantly different from the MK model, the sensor’s measurement performance can be identified as inconsistent with other normal cuffless BPMs. Therefore, the pulse wave simulator and MLR model calibration method proposed in this study can rapidly validate cuffless BPMs and quantify the bias values for further performance fine-tuning.

### 3.2. Limitations and Future Works

The pulse wave simulator designed in this study has two key aspects that could be investigated in future studies. The first is the arterial phantom used in this work; human tissue may have different Young’s modulus characteristics. For example, the size, stiffness, and placement of setting the simulated artery tube may affect the Young’s modulus characteristics. Therefore, the curve of the MK model we calculated using the parameters of previous studies may deviate from the actual measurement [42,43], and further calibration may be required before measurement. However, even if the materials studied have similar mechanical properties to human arterial tissue, the MK model characteristics of the pulse wave simulator may differ from the theoretical model, which should be investigated in further experiments in the future. The second aspect to consider is our pulse wave simulator, which simulates the hemodynamic characteristics of the pulse wave. This means that the pulse wave simulator we proposed can only be used for cuffless BPM devices based on mechanical sensing. The performance of other cuffless BPMs that use different sensing technology cannot be validated in this simulator. For example, to validate cuffless BPM based on dual PPG [44], the pulse wave simulator must also be able to simulate the optical properties of tissues, which would need to be reflected by the optical properties of the skin phantom and arterial phantom, and artificial blood would need to be used as the fluid in the simulator’s circulatory system. In the future, based on further research into these two issues, the proposed pulse wave simulator based on hemodynamics could be further developed to test the performance of more cuffless BPMs with different technical characteristics. The pulse wave simulator proposed in this study is currently used for cuffless BPM based on local PWV with the sensing principle of mechanical waves, which provides a validation method for the local PWV performance of the cuffless BPM.

## 4. Conclusions

In this study, a pulse wave simulator with hemodynamic properties is proposed. This pulse wave simulator has a similar design to the human arm and can simulate the pulse using a radial artery phantom. For most cuffless BPMs based on hemodynamics, the measurement accuracy of PWV will affect the estimation of MAP, SBP, and DBP. Therefore, cuffless BPM must calibrate PWV (or PTT) before BP estimation. We also design a method to calibrate and quantify the performance of cuffless BPMs for local PWV measurements. Using the MLR model, we can effectively map the PWV generated by the pulse wave simulator and the MK model to objectively evaluate quantitative values for the cuffless BPM as a DUT. The measurement error before calibration for the studied system is 1.7–5.99 mmHg. However, after calibration using the MLR model, the measurement error decreases to 0.14–0.48 mmHg; thus, this method can effectively make the cuffless BPM’s measured performance closer to the MK model’s theoretical values. In the future, using our proposed pulse wave simulator with hemodynamic characteristics and MLR calibration, many rapid, quantitative tests can be performed in an industrial production context on cuffless BPMs to identify inaccuracies. Given the forecasted great demand for cuffless BPM technology in home care devices in the future, this study provides a cuffless BPM verification method to ensure reliable measurement performance, thus ensuring consistent performance when used by patients and reducing the risk of device recalls by regulatory agencies.

## Figures and Tables

**Figure 1 micromachines-14-01218-f001:**
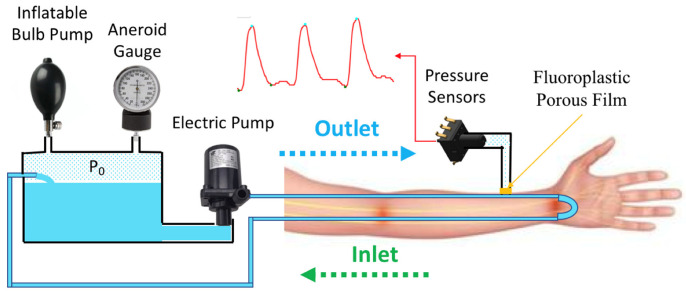
System architecture diagram of the pulse wave simulator.

**Figure 2 micromachines-14-01218-f002:**
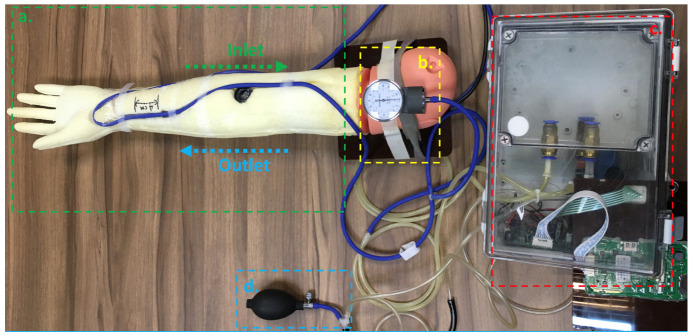
Implementation of the proposed pulse wave simulator. (**a**) An adult-size arm model wears the cuffless BPM for testing; (**b**) The system’s aneroid gauge; (**c**) The printed circuit board (PCB) of the simulator’s control system, the electric pump, and the water reservoir; (**d**) The setting a base pressure.

**Figure 3 micromachines-14-01218-f003:**
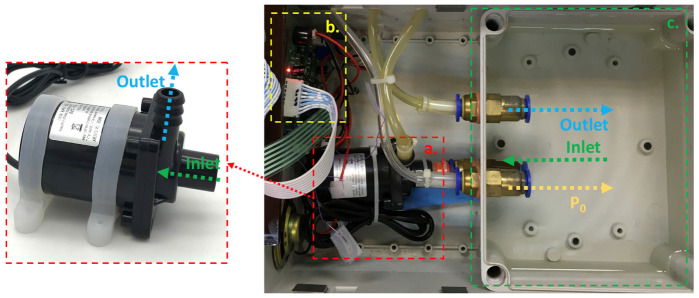
The control system and water reservoir. (**a**) The electric pump; (**b**) The control system’s PCB; (**c**) the water reservoir used to store water in the simulated circulation system.

**Figure 4 micromachines-14-01218-f004:**
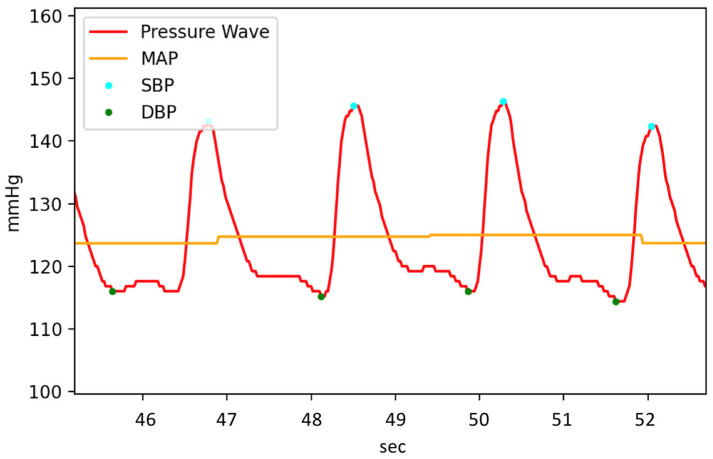
Plot showing the simulated blood pressure (BP) inside an arterial phantom.

**Figure 5 micromachines-14-01218-f005:**
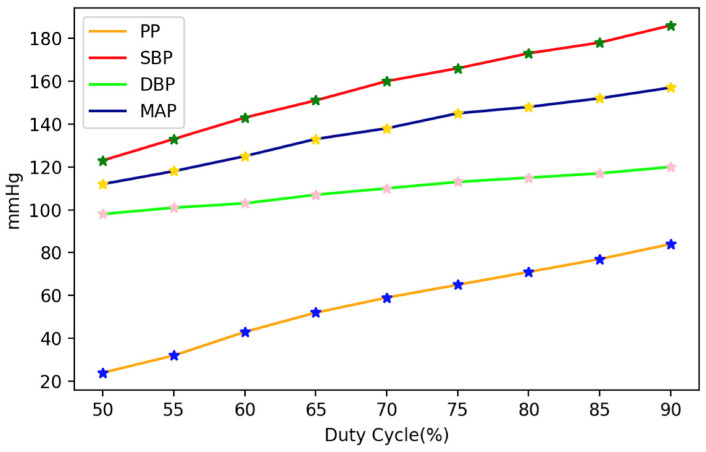
Plot illustrating the BP components at different duty cycle set points.

**Figure 6 micromachines-14-01218-f006:**
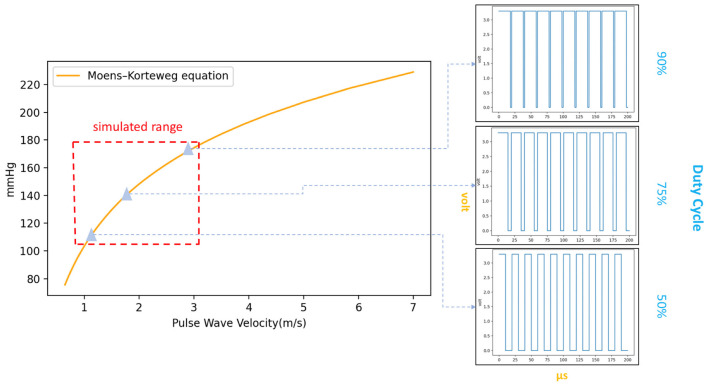
The MK equation curve mapped to different duty cycles.

**Figure 7 micromachines-14-01218-f007:**
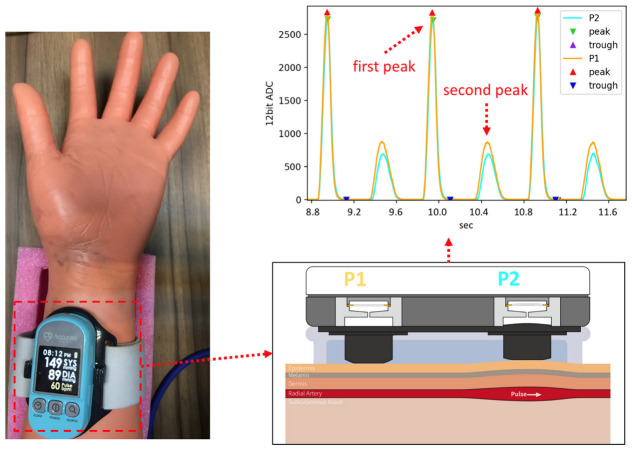
The cuffless BPM used to measure the pulse wave simulator.

**Figure 8 micromachines-14-01218-f008:**
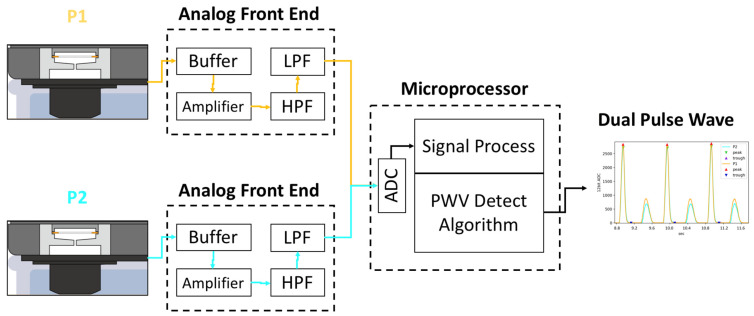
The analog front-end block diagram of the pulse wave sensor.

**Figure 9 micromachines-14-01218-f009:**
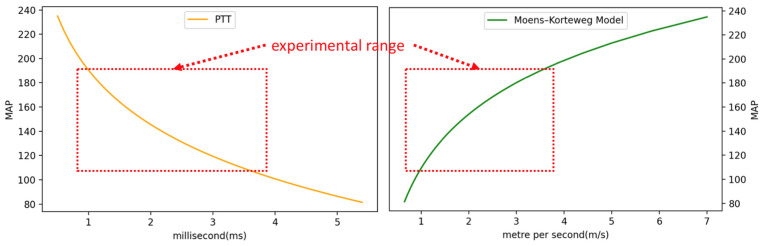
The ranges of PWV and PTT mapped to simulated BP.

**Figure 10 micromachines-14-01218-f010:**
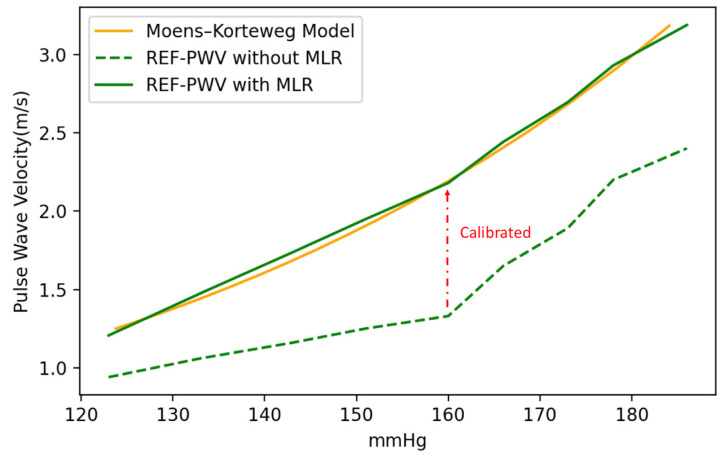
The calibrated PWV measurements using the MLR model.

**Figure 11 micromachines-14-01218-f011:**
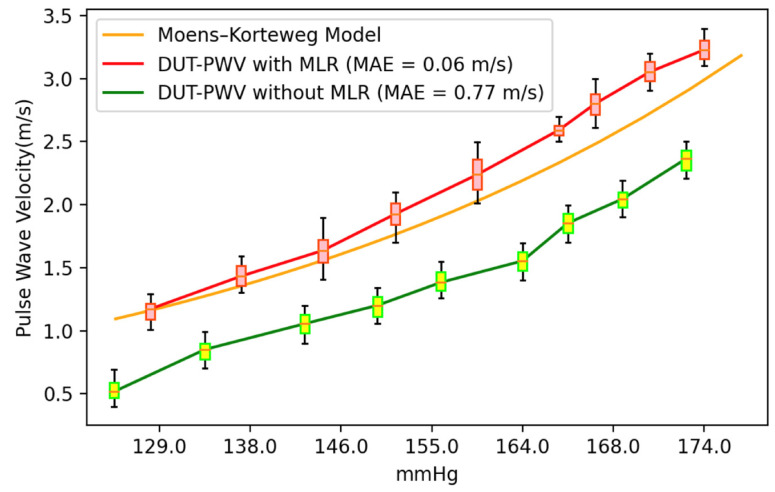
PWV measurement results for the DUT before and after calibration using MLR.

**Figure 12 micromachines-14-01218-f012:**
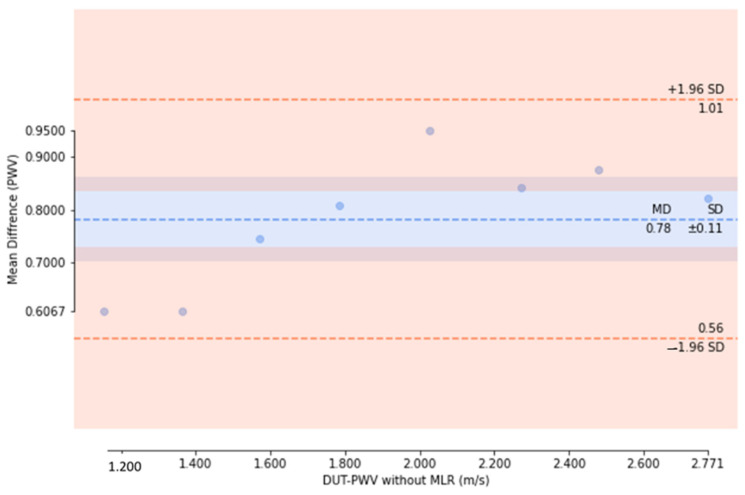
Bland–Altman plot showing PWV measurements before calibration using the MLR model.

**Figure 13 micromachines-14-01218-f013:**
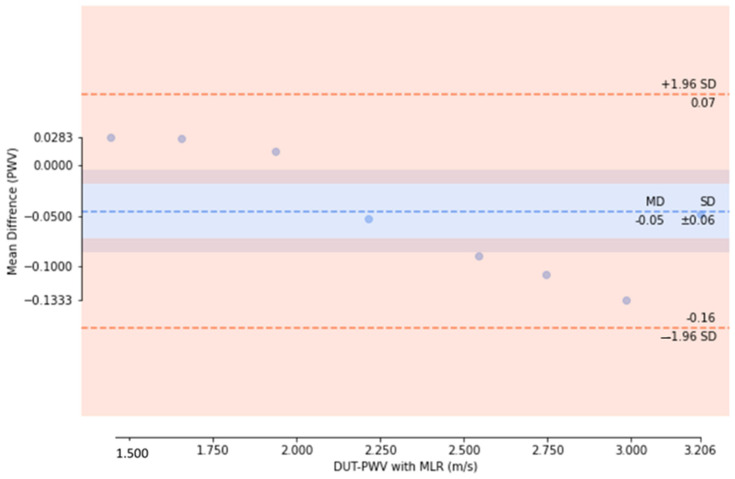
Bland–Altman plot showing PWV measurements after calibration using the MLR model.

## Data Availability

Not applicable.

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
