# Peer review of "A Hemodynamic Pulse Wave Simulator Designed for Calibration of Local Pulse Wave Velocities Measurement for Cuffless Techniques"

_micromachines, 2023, doi:10.3390/mi14061218_

Round 1

Reviewer 1 Report

Please, look the attachement.

Reviewer 2 Report

This manuscript reported a method to develop a pulse wave simulator capable of simulating the human cardiovascular system in vitro including complex hemodynamic phenomena. The authors also aimed to design a cuffless BPM for local PTT measurement and discussed the simulated pulse waves generated in this research. Nonetheless, more explains are also needed to strengthen authors' statements before accepting the manuscript for publication.

1. The formation of arterial pressure is influenced by various factors, including blood filling, cardiac ejection, and the elastic properties of large arteries. In this paper, the pulse wave simulator primarily focuses on simulating blood pressure and does not provide sufficient data support for the elastic characteristics of the silicone hose.

2. The pressure values of the simulated pulse wave depicted range from 115 mmHg to 145 mmHg (Figure 4), which is higher than the typical blood pressure range of 60 mmHg to 120 mmHg in normal individuals. The specific reason and significance behind this discrepancy are not thoroughly explained in the paper. 

3. Normal heart regulation involves both contraction and relaxation, resulting in a pulse waveform that typically consists of two distinct peaks. However, in the simulated waveform presented in the study, only a single peak is observed. Whether the obtained results can be effectively used to construct a calibration model. Further investigation and analysis are required to determine the reasons behind the missing second peak and to explore potential solutions for improving the simulation accuracy.

4. What do the numbers on the abscissa in Figure 9 represent?

-
